# Treatment of Trochanteric Hip Fractures with Cephalomedullary Nails: Single Head Screw vs. Dual Integrated Compression Screw Systems

**DOI:** 10.3390/jcm12103411

**Published:** 2023-05-11

**Authors:** Marye M. Méndez-Ojeda, Alejandro Herrera-Rodríguez, Nuria Álvarez-Benito, Himar González-Pacheco, Miguel A. García-Bello, Javier Álvarez-de la Cruz, José L. Pais-Brito

**Affiliations:** 1Orthopedic Surgery and Traumatology Service, University Hospital of the Canary Islands, 38320 La Laguna, Spain; maryemerce@gmail.com (M.M.M.-O.);; 2Orthopaedic Surgery, Health Sciences, Medicine, La Laguna University, 38320 La Laguna, Spain; 3Canary Islands Health Research Institute Foundation (FIISC), 38109 Tenerife, Spain; 4Network for Research on Chronicity, Primary Care, and Health Promotion, (RICAPPS), Spain

**Keywords:** hip fracture, trochanteric fracture, Gamma nail, InterTAN nail

## Abstract

Extracapsular hip fractures are very common in the elderly. They are mainly treated surgically with an intramedullary nail. Nowadays, both endomedullary hip nails with single cephalic screw systems and interlocking double screw systems are available on the market. The latter are supposed to increase rotational stability and therefore decrease the risk of collapse and cut-out. A retrospective cohort study was carried out, in which 387 patients with extracapsular hip fracture undergoing internal fixation with an intramedullary nail were included to study the occurrence of complications and reoperations. Of the 387 patients, 69% received a single head screw nail and 31% received a dual integrated compression screw nail. The median follow-up was 1.1 years, and in that time, a total of 17 reoperations were performed (4.2%; 2.1% for single head screw nails vs. 8.7% for double head screws). According to the multivariate logistic regression model adjusted for age, sex and basicervical fracture, the adjusted hazard risk of reoperation required was 3.6 times greater when using double interlocking screw systems (*p* = 0.017). A propensity scores analysis confirmed this finding. In conclusion, despite the potential benefits of using two interlocking head screw systems and the increased risk of reoperation in our single center, we encourage to other researchers to explore this question in a wider multicenter study.

## 1. Introduction

Hip fractures (HF) are a frequent problem in elderly patients, and are related to osteopenia and osteoporosis. Around 1.6 million patients suffer from HF per year [1], and by 2050, the global incidence is believed to become 4.5 million [2]. Reduced bone density, female sex (female/male ratio greater than 2/1 in those over 50 years of age), low weight and reduced physical activity are main risk factors for HF [1,2]. The mortality rate among patients who suffer a hip fracture is 5–10% one month after the fracture and 20–30% in the first year [1,2].

Hip fractures can be classified as intracapsular or extracapsular, and the latter are subdivided into basicervical, intertrochanteric/pertrochanteric and subtrochanteric. Up to half of such fractures are intertrochanteric, usually occurring in elderly patients as a result of low-energy trauma [3]. The main treatment of these trochanteric fractures is surgery, which can be extramedullary or intramedullary. Previously, extramedullary treatment with sliding hip screw (SHS) was the most indicated, but some studies showed that nailing gave better fracture fixation results for uncommon trochanteric fractures, especially subtrochanteric fractures, so the use of nailing has dramatically increased even though there is no evidence that it is superior to the SHS in a simple intertrochanteric pattern [3]. 

Some of the most-used intramedullary nailing alternatives in our region are the Gamma3 nail, PFNA (Proximal Femoral Nail Antirotation) and TRIGEN InterTAN. The latter differs from the others in that it offers the possibility of two cephalo-cervical screws that provide linear compression and additional resistance to the rotation of the femoral head, while the others use a single screw [4,5]. Several studies and meta-analyses have compared the use of these methods of treatment without reaching a definitive conclusion on which fixation method is most appropriate to reduce complications and improve prognosis [6,7,8,9,10,11,12,13,14,15,16,17]. Independent risk factors for early mortality already reported in the literature are: male sex, dependence on others for the basic activities of daily living, American Society of Anesthesiologists (ASA) score > 2, older age and medical complications occurring while an inpatient [18]. Fracture stability also plays a key role in the prognosis for these patients, not only influencing early device failure requiring reoperation within 12 months, but also increasing the rate of mortality after trochanteric hip fractures by up to 1.6 times [19]. Studies such as Chehade et al. also describe an increase in early osteosynthesis failure associated with the use of double lag screw systems [19]. Attending to only unstable hip fractures, such as subtrochanteric fractures, Panteli et al. identified six risk factors associated with reoperation: age < 75 years old, pre-injury femoral neck shaft angle, choice of nail, varus reduction angle, fracture-related infection and non-union. The addition of a proximal anti-rotation screw did not confer any benefit [20] in terms of reoperation or survival rates. Figure 1 and Figure 2 show typical cases of pertrochanteric hip fracture treated with an endomedullary nail.

For a time, in our service, we had the impression that systems with two cephalo-cervical screws had a higher rate of reoperations. As such, the aim of this study is to retrospectively review major post-surgical complications as they relate to the type of nail used, comparing single head screw nails to dual integrated compression screw nails, with nails being implanted at a third-level hospital. The hypothesis of our study is that cephalomedullary nailing with double head screw systems present a greater number of post-surgical complications compared to single screw systems.

## 2. Materials and Methods

### 2.1. Study Design

Single-center retrospective cohort study.

### 2.2. Study Subjects

We collected data of patients who underwent surgery and received a single or double head screw hip nail at the Hospital Universitario de Canarias between 1 April 2019 and 26 July 2021 Inclusion Criteria: patients who received a single or double head screw intramedullary hip nail, both short and long. All patients included began rehabilitation treatment with full weight-bearing authorized in the first 24 h after the procedure, as long as the postoperative hemoglobin level was greater than 8 g/dL in the postoperative analysis. Exclusion Criteria: absence of follow-up, use of another fixation system other than the Gamma3 or TRIGEN InterTAN nail, shaft femoral fractures, patients who underwent surgery more than 72 h after suffering the fracture.

In our service, all pertrochanteric, basicervical or subtrochanteric fractures are treated with endomedullary hip nails. Both Gamma3 and InterTAN nails are available in stock, and the choice of implant is at the discretion of the attending physician. Both short and long nails were included. Short nails have been selected for stable fracture patterns (AO/OTA classification 31.A2/31.A3) and long nails for unstable fracture patterns (31.A3). Fracture patterns classified as 31.A2 were treated at the surgeon’s discretion according to his clinical judgment.

### 2.3. Study Variables

Age, sex (female/male), laterality (right/left), type of fracture (AO classification), date of surgery, type of nail (Gamma3/TRIGEN InterTAN), reoperation (yes/no; considering only the first reoperation and the time until first reoperation), reason for reoperation (cut-out/implant failure/infection/others), date of reoperation, exitus and date of exitus.

### 2.4. Data Collection

After approval by the Ethics Committee of the Hospital Universitario de Canarias (code CHUC_2021_134), all patients who met the inclusion criteria and none of the exclusion criteria were identified. A Microsoft Excel-type document was prepared in which patient data related to all study variables were collected, excluding patients’ personal data. All patient medical records were reviewed through the computer system of University Hospital of the Canary Islands to complete the document, and the following data were collected: evolution of hospitalization, discharge reports, follow-up in outpatient consultations.

### 2.5. Statistical Analysis

A descriptive analysis of the sample was made, where the continuous variables were expressed by means and standard deviations (SD), and categorical variables expressed through frequencies and percentages. In addition, a bivariate analysis was carried out using the t-student or chi-square test according to the nature of the variables (continuous or categorical, respectively).

In order to compare post-surgical complications requiring re-intervention, a survival cox regression model was applied. The dependent variable in the model was a dichotomous variable indicating whether reoperation was required (yes/no) and the time until reoperation. The following covariates were included in the model: type of nail (Gamma3/InterTAN), type of fracture (Basicervical/Other), age and sex. Additionally, due to concerns related to the rule of ten outcome events per predictor variable, the effect was also estimated using full propensity score matching with the MatchIt package, which is particularly suitable for modeling rare events.

## 3. Results

A total of 387 medical records were analyzed. The Gamma3 nail was used in 262 patients (67.7%) and InterTAN was used in 125 patients (32.3%). The mean age of the study population was 81.6 (SD = 11.3), of which 74.2% were women (Table 1). The patients were followed up with for a median of 1.1 years, with a maximum follow-up of 2.6 years.

A comparison of the sample based on the type of nail revealed that the two groups had similar characteristics, with no significant differences in age, sex, laterality, or mortality. However, there were some variations in the type of fracture observed. In the InterTAN group, 25.6% of the fractures were basicervical, while 7.3% of the Gamma3 nail group had basicervical fractures (*p* < 0.001). Conversely, the Gamma3 nail group had a higher percentage of subtrochanteric fractures (14.1%) compared to the InterTAN group (5.6%; *p* = 0.022). There was no significant difference in the percentage of intertrochanteric fractures between the two groups (*p* = 0.116) (Table 1).

A total of 17 fractures required reoperation, as shown in Table 2. The reoperation rates were higher in InterTAN group compared to the Gamma3 group (*p* = 0.009). Analysis indicates that cut-out may have been a contributing factor to the difference in reoperation rates between the two groups (*p* = 0.016).

Analyzing which other factor could explain the reoperation rate, we can see there was no significant different rate associated with the side of the fracture (*p* = 0.80), sex (*p* = 0.50) or age (*p* = 0.45).

### 3.1. Risk of Reoperation in the Follow-Up Adjusting by Cox Regression Modeling

The Cox regression model was used to analyze the association between the risk of reoperation and different covariates. The results of the model indicate that the type of nail (Gamma3/InterTAN) was a significant predictor of reoperation, with patients who received the InterTAN having a higher hazard ratio (HR) of 3.6 (95% CI: 1.3, 10.5) for required reoperation compared to those who received the Gamma3 nail. The results of Model 2 (Cox regression model for risk of reoperation needed) indicate that the type of nail (Gamma3 vs. InterTAN) was a significative predictor of reoperation required (Table 3), but neither the type of fracture (basicervical/other) nor age nor sex were found to be significant predictors. These results suggest that the type of nail may be an important factor in determining the risk of reoperation after surgery.

In addition to the Cox regression analysis, a Kaplan-Meier survival curve was generated to visualize the probability of required reoperation over time for the different nails. The results of the Kaplan-Meier analysis show that the probability of required reoperation was higher for patients who received the InterTAN (Figure 3a). The curve for the InterTAN group drops more steeply than the curve for the Gamma3 nail group, indicating that reoperations required were more likely to occur early on, and at a higher rate in the InterTAN group. We can see that the difference was important from one year of follow-up. Additionally, the log-rank test was performed to test the equality of the survival curves between the two groups, with a *p* = 0.001 indicating a statistically significant difference between the groups. Overall, the Kaplan-Meier analysis provides a visual representation of the reoperation required rates, and supports the findings from the Cox regression analysis. Otherwise, we analyzed all-cause mortality and didn’t find any difference between nail groups related to overall survival, *p* = 0.70 (Figure 3b).

### 3.2. Risk of Reoperation Evaluated after Propensity Score Matching

We conducted a propensity score matching, which shows how the balance of the baseline covariates between the treatment groups was assessed to evaluate the success of the matching procedure. The results after propensity score matching show that the distribution of the baseline characteristics, including age, sex and type of fracture, were similar between the treatment groups, with a standardized mean difference of less than 0.1 for all covariates (Figure 4). This suggests that the propensity score matching procedure was successful in controlling for potential confounding effects of the baseline covariates on the treatment effect. Additionally, the effect of the treatment on the outcome of interest, reoperation, was found to be consistent with the results obtained before propensity score matching (HR = 3.3; *p* = 0.038).

## 4. Discussion

The present study showed that in this cohort, double interlocking head screw nailing systems such as the InterTAN nail led to a significantly higher rate of reoperations compared to the Gamma3 nail. At first, it seemed that this finding could be associated with the fracture pattern, as there was a heterogenous distribution of types of fracture. However, no significant differences were found in any other indicators between the two groups, including type of fracture, sex or age.

Although there is some scientific biomechanical proof of the rotational stability of InterTAN nails [5], in clinical studies and reviews, there is still controversy between the existing types of nails [6,7,8,9,10,11,12,13,14,15,16,17]. A priori, this increase in rotational stability would be advantageous in basicervical fracture traces by avoiding rotation of the cervical neck when drilling or inserting the cephalic screw. However, this supposed biomechanical advantage was not reflected in the patients in the study who underwent surgery. 

There have been two Cochrane reviews about trochanteric fracture treatment [3,7]. One compares nails to extramedullary implants [4] and another compares the different types of nails [8]. In the latter review, Queally et al. analyzed 17 randomized clinical trials (RCTs) prior to 2014, compared different nails and concluded that there was insufficient evidence from randomized trials to determine if there are important differences in patient outcomes between the different designs of proximal femoral intramedullary nail produced by different manufacturers when used for the fixation of unstable, or stable, trochanteric fractures [7]. 

There are also more recent clinical trials and meta-analyses comparing different nails and it continues to be uncertain whether there is a difference between implants. Two RCTs specifically compared Gamma3 to InterTAN with similar results [8,9]. Su et al. concluded that no significant difference was found in X-ray times, reduction results, TAD, time to mobilization, operative complications, femoral neck shortening or fracture healing time [8]. Berger et al. affirmed that, in terms of implant-related complications, no significant differences were recorded [9]. Zhang et al. have several studies, including one RCT, comparing InterTAN to PFNA-II in which they didn’t find any significant differences in outcomes except for high pain [10,11,12]. Ülkü et al. retrospectively studied nail migration. Although there was a significant difference in favor of InterTAN, nail migration in the PFNA group did not result in reoperation [13]. Ricci et al. also found a higher radiological collapse in PFNA and DHS vs. InterTAN, but they don’t mention whether that has clinical repercussions [14]. The Liu et al. meta-analysis included two RCTs and seven observational studies, and concluded that patients with the InterTAN nail had a lower risk of screw migration, pain at thigh or hip, cutout, varus collapse of the femoral head, femoral shaft fracture and reoperation. Nonetheless, that finding was based mainly on observational studies, as the researchers didn’t find superiority in cutout, reoperation and femoral shaft fracture when considering only the RCTs [15]. There are two other meta-analyses that suggest that InterTAN leads to fewer complications when compared to single screw devices. However, both of them include mainly retrospective studies and both have conflicts of interest, as they were done by Smith and Nephew collaborators [16,17]. 

Although we have not screened every patient included in the study for osteoporosis, we can affirm that most patients suffered from it to a greater or lesser degree due to their age, comorbidities and the fact that they had suffered the fracture from a low energy impact. A plausible explanation for these results could be the greater aggression to both the head and the femoral neck caused by the integrated double screw. The double reaming performed, coupled with the fact that the double screw system is thicker than the single screw, could further weaken the cortices and the vascularization of an already-weakened bone, increasing the risk of osteosynthesis failure in certain cases.

However, there are several inherent limitations to our study that deserve consideration. First, the retrospective nature presents a potential selection bias. Patients were distributed between treatment groups based on surgeon preference and we didn’t consider the surgeon’s experience in our analysis. Additionally, the pattern of fracture was heterogeneously distributed in both groups and the number of cases is low. Although adjustment was made for several variables, it is possible that residual confounders between the nails could still be present, and therefore the adjusted cox regression and propensity score matching may not be able to adjust or balance all unmeasured confounders. In our center, immediate postoperative radiographs are performed by radiology technicians without the direct supervision and approval of a traumatologist. In several patients, the axial projection of the hip was not performed correctly or was not performed at all. Due of this, it was not possible to perform a correct measurement of the tip-apex distance in all patients, so it was decided not to include it in the study parameters. Lastly, single-center studies lack the external validation required to support changes in practice, so we recommend interpreting these results with caution.

## 5. Conclusions

Single head screw nails such as Gamma3 and dual integrated compression head screw nails such as InterTAN may be effective for surgical treatment of trochanteric fractures. 

A higher risk of reoperation was found when using InterTAN. Therefore, despite the potential biomechanical benefits of using two screws with the InterTAN nail, we cannot recommend it over the Gamma3 nail.

Large-sample multicenter studies may be needed in the future to compare the different cephalomedullary nails available.

## Figures and Tables

**Figure 1 jcm-12-03411-f001:**
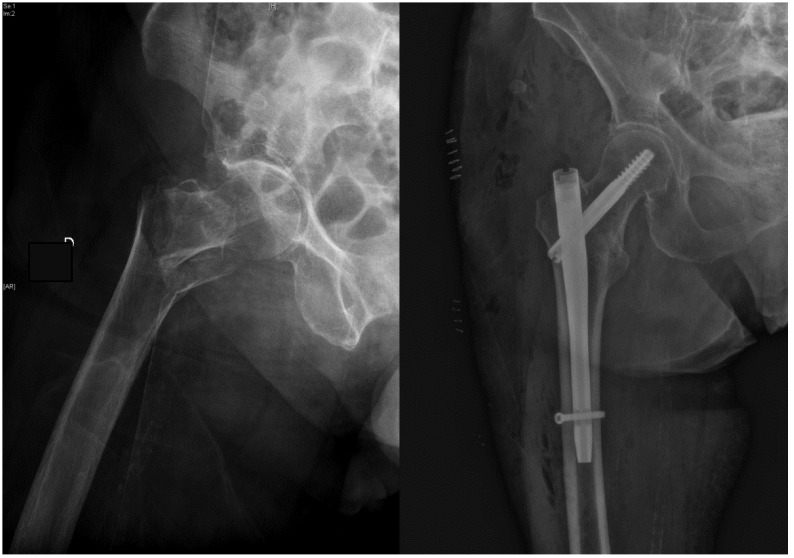
Pertrocantheric hip fracture treated with a short Gamma3 nail.

**Figure 2 jcm-12-03411-f002:**
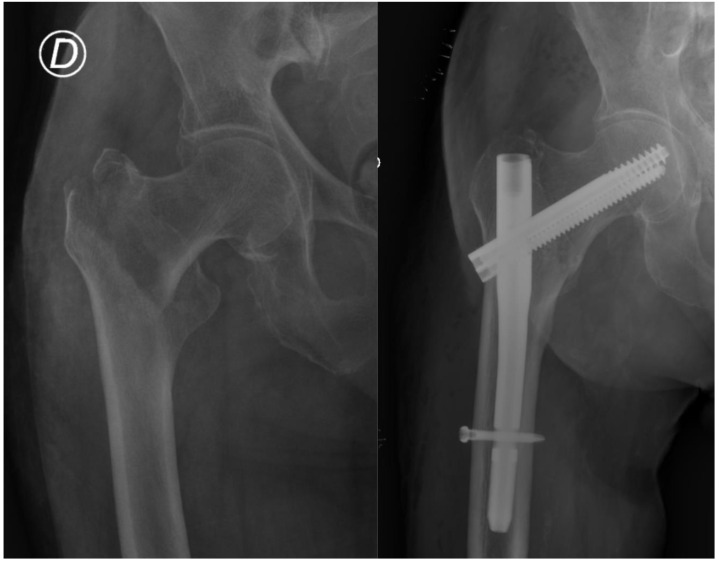
Pertrocantheric hip fracture treated with a short InterTAN nail (letter D for right lower limb).

**Figure 3 jcm-12-03411-f003:**
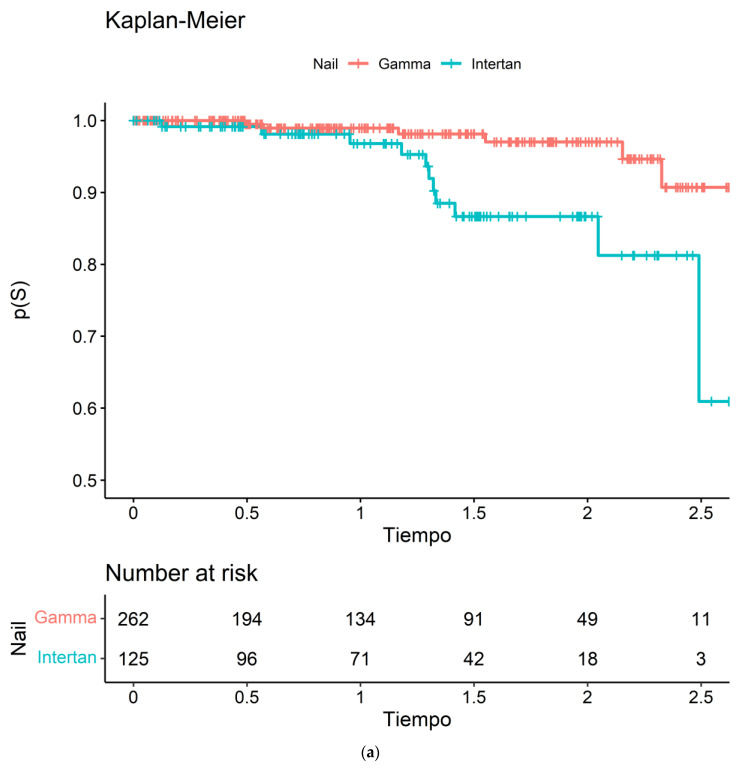
(**a**) Risk of reoperation needed. (**b**) Risk of all-cause mortality.

**Figure 4 jcm-12-03411-f004:**
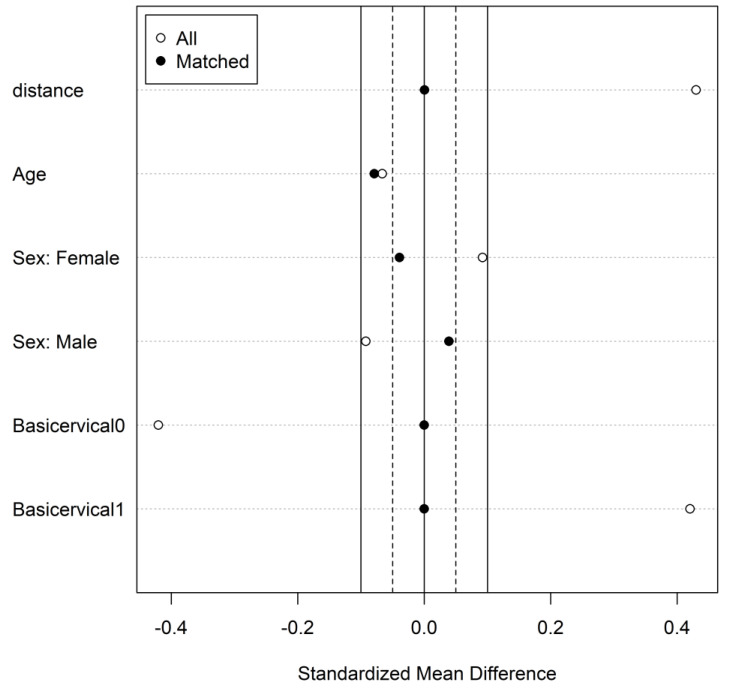
Standardized mean difference of baseline characteristics before and after propensity score matching. The solid vertical line represents the threshold of 0.1, indicating balance between the groups. The matching procedure was successful in balancing the baseline characteristics between the treatment groups.

**Table 1 jcm-12-03411-t001:** Characteristics of the sample according to the type of nail.

	Total (*n* = 387)	Gamma3 (*n* = 262)	InterTAN (*n* = 125)	*p*-Value
Age, mean (SD)	81.6 (11.3)	81.9 (10.7)	81.1 (12.5)	0.524
Gender, Female, *n* (%)	287 (74.2%)	191 (72.9%)	96 (76.8%)	0.487
Laterality, right side, *n* (%)	213 (55.0%)	140 (53.4%)	73 (58.4%)	0.419
Type of fracture, *n* (%)				<0.001
Intertrochanteric	240 (62%)	170 (64.9%)	70 (56%)	0.116
Persubtrochanteric	52 (13.4%)	36 (13.7%)	16 (12.8%)	0.925
Basicervical	51 (13.2%)	19 (7.3%)	32 (25.6%)	<0.001
Subtrochanteric	44 (11.4%)	37 (14.1%)	7 (5.6%)	0.022

SD = Standard Deviation.

**Table 2 jcm-12-03411-t002:** Incidence of complications in the follow-up period after surgery.

	Total (*n* = 387)	Gamma3 (*n* = 262)	InterTAN (*n* = 125)	*p*-Value
Median of follow up (P25–P75)	1.1 (0.5–1.8)	1.0 (0.5–1.8)	1.2 (0.6–1.7)	0.82
Reoperations in the follow-up period after surgery, *n* (%)	17 ^&^ (4.4%)	6 (2.3%)	11 ^&^ (8.8%)	0.008
Reoperation required, rates at 1.5 Years Following Surgery *, %	5.8%	1.9%	13.3%	0.009
Any complication, rates at 1.5 Years Following Surgery, *n* (%)	18 (4.7%)	6 (2.3%)	12 (9.6%)	0.003
Cut-out	8 (2.1%)	2 (0.8%)	6 (4.8%)	0.016
Peri-implant fracture	6 (1.6%)	2 (0.8%)	4 (3.2%)	0.089
Nail Tear	3 (0.8%)	1 (0.4%)	2 (1.6%)	0.245
Infection	1 (0.3%)	1 (0.4%)	0	>0.99
Second reoperation in the follow-up period after surgery, *n* (%)	2 (0.5%)	1 (0.4%)	1 (0.8%)	0.54
Exitus Rates at 1.5 Year Following Surgery *, %	26.9%	26.4%	27.8%	0.79

* Kaplan-Meier curve survival estimations. ^&^ One InterTAN patient required a reoperation, but it was ruled out due to health problems.

**Table 3 jcm-12-03411-t003:** Cox regression model for risk of reoperation needed.

	Model 1: Not Including Type of Nail		Model 2: Including Type of Nail	
	HR [95% CI]	*p*-Value	HR [95% CI]	*p*-Value
InterTAN vs. Gamma3 nail	-	-	3.6 [1.3–10.5]	0.017
Basicervical fracture	2.3 [0.82–6.7]	0.11	1.4 [0.47–4.3]	0.54
Age	0.99 [0.95–1.03]	0.70	1.0 [0.96–1.04]	0.94
Men	1.3 [0.41–4.1]	0.66	1.4 [0.43–4.4]	0.58
McFadden pseudo-R2	0.017		0.052	

## Data Availability

The data presented in this study are openly available in Figshare at https://doi.org/10.6084/m9.figshare.21548340.v2 (accessed on 3 May 2023).

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
