# Peer review of "Treatment of Trochanteric Hip Fractures with Cephalomedullary Nails: Single Head Screw vs. Dual Integrated Compression Screw Systems"

_jcm, 2023, doi:10.3390/jcm12103411_

Round 1
Reviewer 1 Report
ABSTRACT
Please use dot instead of commas when reporting values.
1. INTRODUCTION
Please make sure that the manuscript is read and corrected by a native English-speaker before resubmission. This is very important to ensure that the paper is sent for production in the correct language. This must be done before online publication.
Line 32 - should be "per year".
Line 33 - should be "to become"
Line 35 - remove "the"
Line 51 - should be "the other ones"
Line 54-57 - You need to improve form.
Line 63 - put space after "age"
Line 64 - put "as well" at the end of the sentence.
Line 67-73 this paragraph is very usefull, it clearifies a lot the aim of your paper. Good job
2 MATHERIALS AND METHODS
c. study variables
What about time between fracture and the surgical procedure? was it standard? it is an important factor in fracture surgery and it needs to be clarified
You should write about post-op protocol too.
Line 144 - should be "might be an important factor"
Figure 1a and 1b - please translate in english
Your statistical analysis is great, you need to give it value in your discussion section.
DISCUSSION
Line 184-190 unnecessarily long
Line 194 please correct form; what is TAD? need to explain every abbreviation
Line 196 - Don't start a sentence with "and".
Line 180-213 discussion shouldn't be a "literature review" section, you have to discuss about the findings of your work and critically analyze them. You need to write about the findings of your study and to explain their meaning. You need to express your opinion on your findings: why do you think Gamma3 yields better results? Why were you using InterTAN with basicervical fractures? why do you think that complications are found after just one year? How does this paper change your surgical treatment?
Author Response
ABSTRACT
Please use dot instead of commas when reporting values.
Response: Thank you for the correction. It will be reflected in the corrected article.
1. INTRODUCTION
Please make sure that the manuscript is read and corrected by a native English-speaker before resubmission. This is very important to ensure that the paper is sent for production in the correct language. This must be done before online publication.
Line 32 - should be "per year".
Response: Thank you for the correction. It will be reflected in the corrected article.
Line 33 - should be "to become"
Response: Thank you for the correction. It will be reflected in the corrected article.
Line 35 - remove "the"
Response: Thank you for the correction. It will be reflected in the corrected article.
Line 51 - should be "the other ones"
Response: Thank you for the correction. It will be reflected in the corrected article.
Line 54-57 - You need to improve form.
Response: Thank you for the correction. It will be reflected in the corrected article.
Line 63 - put space after "age"
Response: Thank you for the correction. It will be reflected in the corrected article.
Line 64 - put "as well" at the end of the sentence.
Response: Thank you for the correction. It will be reflected in the corrected article.
Line 67-73 this paragraph is very usefull, it clearifies a lot the aim of your paper. Good job
2 MATHERIALS AND METHODS
c. study variables
What about time between fracture and the surgical procedure? was it standard? it is an important factor in fracture surgery and it needs to be clarified
You should write about post-op protocol too.
Response: All the patients included in the study underwent surgery within 72 hours after suffering the hip fracture. This will be corrected in the exclusion criteria section of the article. In addition, all patients who undergo surgery in our hospital begin rehabilitation treatment with full weight-bearing authorized in the first 24 hours after surgery, as long as the postoperative hemoglobin level was greater than 8g/dL in the postoperative blood test. This will be corrected in the inclusion criteria section of the article.
Line 144 - should be "might be an important factor"
Response: Thank you for the correction. It will be reflected in the corrected article.
Figure 1a and 1b - please translate in english
Response: Thank you for the correction. It will be reflected in the corrected article.
Your statistical analysis is great, you need to give it value in your discussion section.
DISCUSSION
Line 184-190 unnecessarily long
Response: Thank you for the correction. It will be reflected in the corrected article.
Line 194 please correct form; what is TAD? need to explain every abbreviation
Response: Tip-apex distance. Thank you for the correction. It will be reflected in the corrected article.
Line 196 - Don't start a sentence with "and".
Thank you for the correction. It will be reflected in the corrected article.
Line 180-213 discussion shouldn't be a "literature review" section, you have to discuss about the findings of your work and critically analyze them. You need to write about the findings of your study and to explain their meaning. You need to express your opinion on your findings: why do you think Gamma3 yields better results? Why were you using InterTAN with basicervical fractures? why do you think that complications are found after just one year? How does this paper change your surgical treatment?
Response: We really appreciate your suggestions. We will answer that questions in the corrected version of the article.
Although we have not screened every patient included in the study for osteoporosis, we can affirm that most of them suffered from it to a greater or lesser degree, both because of their age and the fact that they had suffered the fracture from a fall from their own height. A plausible explanation for these results could be the greater aggression to both the head and the femoral neck caused by the integrated double screw. The double reaming performed, while the double screw system is thicker than the single screw, could further weaken the cortices and the vascularization of an already weakened bone, increasing in certain cases the risk of osteosynthesis failure.
A priori, this increase in rotational stability would be advantageous in basicervical fracture traces, avoiding rotation of the cervical neck when drilling or inserting the cephalic screw. However, this supposed biomechanical advantage was not reflected in the patients who underwent surgery in the study.
Single-center and retrospective studies lack of external validation required to support changes in practice, so we recommend interpret these results with caution.
Reviewer 2 Report
The paper "Treatment of trochanteric hip fractures with cephalomedullary nails:single head screw vs dual integrated compression screw systems." has been carefully examined and the reviewer feels that the paper needs major revision. The following specific review comments are for the authors' reference.
1. As the type of fracture between two groups are significant different, the Basicervical type was higher in InterTAN group and Subtrochanteric type was higher in Gamma3 group, which may impact the implant failure and reoperation rate, could make bias to the result.
2. The severity of osteoporosis is needed to be addressed as it is a key factor impact implant failure and clinical result of trochanteric hip fractures operation.
3. As the study is a retrospective cohort study, are these two cohort at the same time? What is the philosophy of implant type for the trochanteric hip fractures? How many surgeon enrolled in this study? The type and rate of different implant between different surgeon also needed to be addressed.
Author Response
As the type of fracture between two groups are significant different, the Basicervical type was higher in InterTAN group and Subtrochanteric type was higher in Gamma3 group, which may impact the implant failure and reoperation rate, could make bias to the result.
Response: When the different distribution in terms of fracture patterns was discussed in the study, we decided to use a cox regression model to analyze the association between the risk of reoperation vs fracture patern/type of nail/age/gender. The results of the model indicate that the type of nail (Gamma3 vs. InterTAN) acts in our sample as a significative predictor of reoperation required (Table 3), but neither the type of fracture (Basicervical/Other), nor age, nor gender were found to be significant predictors.
In addition, we conducted a propensity score matching, which shows how the balance of the baseline covariates between the treatment groups was assessed to evaluate the success of the matching procedure. The results after propensity score matching show that the distribution of the baseline characteristics, including age, gender and type of fracture, were similar between the treatment groups, with a standardized mean difference of less than .1 for all covariates (Figure 2).
However, this is a single-center study, with its inherent limitations, so we recommend interpret these results with caution.
The severity of osteoporosis is needed to be addressed as it is a key factor impact implant failure and clinical result of trochanteric hip fractures operation.
Although we have not screened every patient included in the study for osteoporosis, we could estimate that most of them suffered from it to a greater or lesser degree. Both because of their age, other comorbidities, gender and the fact that they had suffered the fracture from a fall from their own height.
Mentioning again item 3.2 of the manuscripts, he results after propensity score matching show that the distribution of the baseline characteristics, including age, gender and type of fracture, were similar between the treatment groups, with a standardized mean difference of less than .1 for all covariates (Figure 2). This suggests that the propensity score matching procedure was successful in controlling for potential confounding effects of the baseline covariates on the treatment effect. Additionally, the effect of the treatment on the outcome of interest, reoperation, was found to be consistent with the results obtained before propensity score matching (HR=3.3; p=.038).
A plausible explanation for these results could be the greater aggression to both the head and the femoral neck caused by the integrated double screw. The double reaming performed, coupled with the fact thatthe double screw system is thicker than the single screw, could further weaken the cortices and the vascularization of an already weakened bone, increasing in certain cases the risk of osteosynthesis failure.
- As the study is a retrospective cohort study, are these two cohort at the same time? What is the philosophy of implant type for the trochanteric hip fractures? How many surgeon enrolled in this study? The type and rate of different implant between different surgeon also needed to be addressed.
Response:
-Both cohorts were conducted at the same time.
-In our service, all pertrochanteric, basicervical or subtrochanteric fractures are treated with endomedullary hip nails. Both GAMMA3 and INTERTAN nails are available in stock, and the choice of implant is at the discretion of the attending physician.
-In principle, we included in the study all patients who underwent surgery within a one-year period and who met the inclusion criteria, regardless of the surgeon. In any case, in our department, hip fractures are scheduled in programmed operating rooms assigned by order to all the attending physicians of the department, with a very similar ratio of hip fractures per surgeon among all.
---------------------------------------------------------------------
However, we have made important changes in both the introduction and discussion of the manuscript, which we would greatly appreciate your review.
Reviewer 3 Report
The authors compared the gamma3 nail to the inter tan nail. They found inter tan nails increases the risk of revision within the first couple of years and confirmed the results with a propensity match. I have a couple of questions for the authors. Given that cutout and peri implant fractures were higher for intertan, why do the authors believe this occurred but was not evident in previous studies? Were all nails in the study long nails or trochanteric nails? Long nails would minimize the likelihood of peri-implant fracture but have additional drawbacks of their own. It’s unlikely a simple double compression screw impacts the likelihood of peri implant fracture. How were the groups compiled? Did the surgeons choose the nail based on comfort (thus introducing surgeon bias) or was it based on a time threshold where only one nail was used at the institution during a certain time period. This is another potential confounder. Did the authors measure tip-apex distance to determine if this was the potential higher cause of failure? This is especially important if different surgeons used different nail types. There are a couple of spelling and grammatical issues that can be cleaned up as well during the revision stages.
Author Response
Given that cutout and peri implant fractures were higher for intertan, why do the authors believe this occurred but was not evident in previous studies?
Although we have not screened every patient included in the study for osteoporosis, we can affirm that most of them suffered from it to a greater or lesser degree, both because of their age and the fact that they had suffered the fracture from a fall from their own height. A plausible explanation for these results could be the greater aggression to both the head and the femoral neck caused by the integrated double screw. The double reaming performed, coupled with the fact that the double screw system is thicker than the single screw, could further weaken the cortices and the vascularization of an already weakened bone, increasing in certain cases the risk of osteosynthesis failure.
However, this is a single-center study, with its inherent limitations, so we recommend interpreting these results with caution.
Were all nails in the study long nails or trochanteric nails? Long nails would minimize the likelihood of peri-implant fracture but have additional drawbacks of their own. It’s unlikely a simple double compression screw impacts the likelihood of peri implant fracture.
Response:
In the study both short and long nails were included. The choice of long or short nails in this type of fractures is still a matter of debate in the literature. The greater biomechanical stability of long nails has as a counterpart a longer intraoperative time and blood loss, so systematizing the use of long nails in fractures that could be adequately synthesized with a short nail is a decision still open to debate. In our center we use short nails for stable fracture patterns ( AO/OTA classification 31.A2/31.A3) and long nails for the unstable ones (31.A3). 31.A2 fracture patterns are treated at the surgeon's discretion according to his clinical judgment.
How were the groups compiled?
We collected data of patients who underwent surgery and who received a single or double head screw hip nail operated at the Hospital Universitario de Canarias between 04.01.2019 and 07.26.2021. Applied the inclusion and exclusion criteria. A total of 387 medical records were analyzed. Then divided the patients into 2 groups, depending on the nail chosen. Gamma3 nail was used in 262 patients and InterTAN was used in 125 patients.
A comparison of the sample based on the type of nail revealed that the two groups had similar characteristics, with no significant differences in age, gender, laterality, or mortality.
We conducted a propensity score matching, which shows how the balance of the baseline covariates between the treatment groups was assessed to evaluate the success of the matching procedure. The results after propensity score matching show that the distribution of the baseline characteristics, including age, gender and type of fracture, were similar between the treatment groups, with a standardized mean difference of less than .1 for all covariates (Figure 2).
This suggests that the propensity score matching procedure was successful in controlling for potential confounding effects of the baseline covariates on the treatment effect. Additionally, the effect of the treatment on the outcome of interest, reoperation, was found to be consistent with the results obtained before propensity score matching (HR=3.3; p=.038).
Did the surgeons choose the nail based on comfort (thus introducing surgeon bias) or was it based on a time threshold where only one nail was used at the institution during a certain time period. This is another potential confounder.
In our service, all pertrochanteric, basicervical or subtrochanteric fractures are treated with endomedullary hip nails. Both GAMMA3 and INTERTAN nails are available in stock, and the choice of implant is at the discretion of the attending physician.
Did the authors measure tip-apex distance to determine if this was the potential higher cause of failure? This is especially important if different surgeons used different nail types.
It was not possible to perform a correct measurement of the tip-apex distance in all patients, so it was decided not to include it in the study parameters. In the immediate postoperative radiographs, there were problems with the axial projection of the hip, many of them being unreliable for accurate measurement. However, and although it is not included in the study either because it lacks clinical relevance, the mean tip-apex distances exclusively in the AP projection are very similar in both groups.
There are a couple of spelling and grammatical issues that can be cleaned up as well during the revision stages.
Thank you for the correction. It will be reflected in the corrected article.
Round 2
Reviewer 3 Report
Although the authors responded to my comments in a point by point fashion this is not reflected in the manuscript. Please add the following comments into the manuscript so it is clear to the reader what your methodology and limitations are:
1) Please add to methodology
Were all nails in the study long nails or trochanteric nails? Long nails would minimize the likelihood of peri-implant fracture but have additional drawbacks of their own. It’s unlikely a simple double compression screw impacts the likelihood of peri implant fracture.
Response:
In the study both short and long nails were included. The choice of long or short nails in this type of fractures is still a matter of debate in the literature. In our center we use short nails for stable fracture patterns ( AO/OTA classification 31.A2/31.A3) and long nails for the unstable ones (31.A3). 31.A2 fracture patterns are treated at the surgeon's discretion according to his clinical judgment.
2) Please add to methodology
Did the surgeons choose the nail based on comfort (thus introducing surgeon bias) or was it based on a time threshold where only one nail was used at the institution during a certain time period. This is another potential confounder.
In our service, all pertrochanteric, basicervical or subtrochanteric fractures are treated with endomedullary hip nails. Both GAMMA3 and INTERTAN nails are available in stock, and the choice of implant is at the discretion of the attending physician.
3) Please add to limitations section
Did the authors measure tip-apex distance to determine if this was the potential higher cause of failure? This is especially important if different surgeons used different nail types.
It was not possible to perform a correct measurement of the tip-apex distance in all patients, so it was decided not to include it in the study parameters. In the immediate postoperative radiographs, there were problems with the axial projection of the hip, many of them being unreliable for accurate measurement. However, and although it is not included in the study either because it lacks clinical relevance, the mean tip-apex distances exclusively in the AP projection are very similar in both groups.
Author Response
Thank you for your corrections. We have added in the latest version of the manuscript the clarifications suggested in the methodology and discussion.